# Active School Transport among Children from Canada, Colombia, Finland, South Africa, and the United States: A Tale of Two Journeys

**DOI:** 10.3390/ijerph17113847

**Published:** 2020-05-28

**Authors:** Silvia A. González, Olga L. Sarmiento, Pablo D. Lemoine, Richard Larouche, Jose D. Meisel, Mark S. Tremblay, Melisa Naranjo, Stephanie T. Broyles, Mikael Fogelholm, Gustavo A. Holguin, Estelle V. Lambert, Peter T. Katzmarzyk

**Affiliations:** 1School of Medicine, Universidad de los Andes, Bogota 111711, Colombia; osarmien@uniandes.edu.co (O.L.S.); ms.naranjo10@uniandes.edu.co (M.N.); ga.holguin@uniandes.edu.co (G.A.H.); 2Healthy Active Living and Obesity Research Group, Children’s Hospital of Eastern Ontario Research Institute, Ottawa, ON K1H 8L1, Canada; richard.larouche@uleth.ca (R.L.); mtremblay@cheo.on.ca (M.S.T.); 3School of Epidemiology and Public Health, Faculty of Medicine, University of Ottawa, Ottawa, ON, K1G 5Z3, Canada; 4Centro Nacional de Consultoría, Bogota 110221, Colombia; plemoine@cnccol.com; 5Faculty of Health Sciences, University of Lethbridge, 4401 University Drive, Lethbridge, AB, T1K 3M4, Canada; 6Facultad de Ingeniería, Universidad de Ibagué, Ibagué 730001, Colombia; jd.meisel28@uniandes.edu.co; 7Pennington Biomedical Research Center, Baton Rouge, LA 70808, USA; stephanie.broyles@pbrc.edu (S.T.B.); peter.katzmarzyk@pbrc.edu (P.T.K.); 8Department of Food and Nutrition, University of Helsinki, 00100 Helsinki, Finland; mikael.fogelholm@helsinki.fi; 9Research Centre for Health through Physical Activity, Lifestyle and Sport, Division of Exercise Science and Sports Medicine, Faculty of Health Sciences, University of Cape Town, Rondebosch, Cape Town 7700, South Africa; vicki.lambert@uct.ac.za

**Keywords:** active school transport, distance, safety, Canada, Colombia, Finland, South Africa, United States

## Abstract

Walking and biking to school represent a source of regular daily physical activity (PA). The objectives of this paper are to determine the associations of distance to school, crime safety, and socioeconomic variables with active school transport (AST) among children from five culturally and socioeconomically different country sites and to describe the main policies related to AST in those country sites. The analytical sample included 2845 children aged 9–11 years from the International Study of Childhood Obesity, Lifestyle and the Environment. Multilevel generalized linear mixed models were used to estimate the associations between distance, safety and socioeconomic variables, and the odds of engaging in AST. Greater distance to school and vehicle ownership were associated with a lower likelihood of engaging in AST in sites in upper-middle- and high-income countries. Crime perception was negatively associated to AST only in sites in high-income countries. Our results suggest that distance to school is a consistent correlate of AST in different contexts. Our findings regarding crime perception support a need vs. choice framework, indicating that AST may be the only commuting choice for many children from the study sites in upper-middle-income countries, despite the high perception of crime.

## 1. Introduction

In the context of a global crisis of physical inactivity, walking and biking to and from school represent an opportunity for children to engage in regular physical activity (PA) on a daily basis [1]. Children who walk or bike to school have higher levels of PA [2] and lower measures of adiposity [3,4]. Moreover, cycling to/from school is associated with higher cardiorespiratory fitness [3] and lower cardiovascular disease risk factors [5]. In addition to the health benefits, active school transport (AST) contributes to the development of children’s independent mobility [6], provides opportunities for children to interact with their local environments [7], and has the potential to mitigate the adverse environmental effects of the use of motorized vehicles around schools by reducing emissions of greenhouse gases and other pollutants [8,9]. Despite these benefits, time trends show that the prevalence of AST in low, middle- [10,11,12,13], and high-income countries (HIC) [14,15,16,17,18,19,20,21,22] is declining.

A multicountry study indicated that more than 50% of children reported AST in cities from upper-middle-income countries (UMIC) like Bogota in Colombia and Cape Town in South Africa and in cities from HIC like Helsinki in Finland and Bath in the UK [2]. These findings can be understood, in part, within a “need vs. choice” framework, where the drivers to engage in AST differ according to the context [23]. This framework proposes that for low and middle-income countries (LMIC), the involvement in AST may be a result of a need given the limited car availability [4,23]. In contrast, for HIC, AST could be a choice driven by the availability of policies and infrastructure that support AST [4,24]. This framework suggests that the drivers for AST may differ according to the context and taking those differences into account is essential for policymaking processes. 

Although previous studies have shown associations between AST and several correlates, including distance, motor vehicle ownership, perceived safety, land use mix, walking and cycling infrastructure, walkability, urban form, and social interactions [25,26,27,28,29], there is evidence that shows that associations and the direction of association may differ across countries [25]. Furthermore, distance between home and school has been described as the most consistent correlate of AST [26,27,30], and together with safety perceptions and resources availability, such as car ownership, are among the main factors that influence AST and can guide public policy design [31]. However, the studies that have objectively measured the distance to school have been conducted only in HIC [30]. Therefore, the generalizability of these results to UMIC and the relevance of these factors in the design of AST policies is unclear.

In this context, international studies using comparable methods and including sites that differ on sociodemographic characteristics can help to elucidate the association between AST with potential correlates such as distance and crime safety. Furthermore, a review of the policy environment in these contexts is crucial to determine the relevance of environmental variables like distance and safety in the practice to promote AST. Therefore, the objectives of this study are twofold: (1) to determine the associations between AST and measures of distance between home and school and crime safety among children from study sites located in five culturally and socioeconomic different countries and (2) to describe the main policies related to AST in the research sites included in this study. 

## 2. Materials and Methods

The International Study of Childhood Obesity, Lifestyle and the Environment (ISCOLE) is a multinational, cross-sectional study conducted among 9–11-year-old children from study sites in 12 countries. More details on the study design and methods can be found elsewhere [32]. Analyses reported in the present study include data from five ISCOLE sites: Ottawa (Canada), Bogota (Colombia), Helsinki, Espoo, and Vantaa; onwards, we will use Helsinki as a collective term for the three cities (Finland), Cape Town (South Africa), and Baton Rouge (United States). Data were collected in 111 schools (Ottawa = 26, Bogota = 20, Helsinki = 25, Cape Town = 19 and Baton Rouge = 21). These sites were included in the present analyses because they provided objective data on distance between home and school measured using geographic information systems (GIS). 

The institutional review board at the Pennington Biomedical Research Center (coordinating center) approved the overarching protocol, and the institutional/ethical review boards at each participating institution approved local protocols. Written informed consent was obtained from parents or legal guardians, and child assent was obtained for all participants. The data were collected from September 2011 through December 2013.

### 2.1. Study Setting

Our study included sites in the countries with the most unequal (South Africa) and the least unequal (Finland) distribution of income in ISCOLE, according to the Gini index [33]. The population at the city levels varied from 812,129 inhabitants in Ottawa to 7,674,366 inhabitants in Bogota [34]. Some variability was also observed in contextual variables at the national level that may be relevant to engagement in AST. The number of motor vehicles per capita ranged from 89 per 1000 inhabitants in Colombia, to 809 per 1000 inhabitants in the United States [35]. The road traffic death rate ranged from 5.1 to 31.9 deaths per 100,000 inhabitants, in Finland and South Africa, respectively [36], and robbery rate varied from 30.8 to 197.5 per 100,000 inhabitants, in Finland and Colombia, respectively [37] (Table 1).

### 2.2. Participants

The overall response rate for ISCOLE study was 60% [38]. The present analysis included 2960 children from the selected study sites, and 2845 remained in the analytical data set after excluding participants for whom data on AST (*n* = 22), parental education (*n* = 17), motor-vehicle availability (*n* = 9), and parental crime perception (*n* = 67) were not available. The inclusion rate per study site was 95.4% for Ottawa, 99.2% for Bogota, 90.8% for Helsinki, 56.7% for Cape Town, and 91% for Baton Rouge. The participants who were excluded from the present analysis were more likely to be overweight or obese (*p* < 0.001) and were less likely to report meeting PA guidelines (*p* = 0.001), compared with the included sample. In addition, their parents were more likely to report that they did not complete high school (*p* < 0.001). The five study sites were specifically selected from the 12 country sites for the ISCOLE study on the basis of the availability of geo-coded data.

### 2.3. Measurements

#### 2.3.1. Active School Transport

AST was self-reported by participants, who responded to a diet and lifestyle questionnaire [32]. The questions used to assess AST were adapted for each country from the Canadian component of the 2009–2010 Health Behavior in School-aged Children Study [39]. Travel mode was assessed with the question “In the last week you were in school, the main part of your journey to school was by”. The response options included active modes such as walking, bicycle, roller blades, and scooter and motorized modes such as car, motorcycle, moped (motor scooters), bus, train, tram, underground or boat, and others according to country-specific modes of transport. Other modes of transportation included active modes such as running and jogging; motorized modes such as the school van, bus feeder; and inactive nonmotorized modes such as pedicab (tricycle with a passenger compartment), and riding on the top tube of the bike’s frame [4]. For this analysis, responses were collapsed into a binary variable indicating AST or nonactive travel. 

#### 2.3.2. Distance to School

Distance to school was estimated using ArcGIS 10.2 (ESRI Inc., Redlands, CA). Children’s home address information was reported by the parents who responded to a demographic questionnaire. Home and school addresses were geo-coded using specific layers for each city. If parents did not provide a complete address, the closest street intersection was used. To estimate the distance between home and school, it was assumed that children took the shortest route via the street network. For the present analysis, distance was used both as a continuous and as a categorical variable using 4 levels: (1) <1000 m; (2) 1000 m–1499 m; (3) 1500 m–1999 m; and (4) ≥2000 m. The categories of distance to school were determined by previous studies and examining the continuous measurement of distance to school using smoothed locally estimated scatterplot smoothing (LOESS) curves [40].

#### 2.3.3. Parental Perception of Crime

We included a crime perception scale adapted from the Neighborhood Environment Walkability Scale for Youth (NEWS-Y) [41], created based on five crime safety items and assessed on a 4-point Likert scale (from strongly disagree to strongly agree): “I am afraid of my child being taken or hurt by a stranger on local streets”, “I am afraid of my child being taken or hurt by a stranger in my yard, driveway or common area”, “I am afraid of my child being taken or hurt by a stranger in a local park”, “I am afraid of my child being taken or hurt by a known ‘bad’ person (adult or child) in my neighborhood”, and “there is a high crime rate”. The crime perception variable was scored (score range 1–4) as the average of the five crime safety items (Cronbach α = 0.86) [25], where high scores represent greater safety concerns. 

#### 2.3.4. Correlates

Sociodemographic variables were reported by the parents in response to the demographic and family health questionnaire from ISCOLE [32]. For this analysis we included age, gender, parental education, and vehicle ownership. The highest parental education variable was created based on the highest education level attained by the mother or the father (less than high school, complete high school or some college, and university degree or postgraduate degree). Vehicle ownership was reported as the number of motorized vehicles (cars, motorcycles, mopeds, and/or trucks) available for use in the household and was recoded as 0 vs.1 vs. ≥ 2 for the analyses.

### 2.4. Statistical Analysis

The descriptive characteristics included the means and standard deviations (SD) for continuous variables and the frequencies of categorical variables stratified by study site. Associations between distance to school and the likelihood of engaging in AST were estimated using generalized linear mixed models (SAS PROC GLIMMIX), stratifying by the World Bank country classification by income level of the countries where the study sites were located at, which grouped Bogota and Cape Town as belonging to UMIC and Ottawa, Helsinki, and Baton Rouge as belonging to HIC. The statistical models included age, gender, parental education, motorized vehicle ownership, and crime perception as potential correlates. To account for the clustering effects of schools and study sites, the multilevel models included three levels: the child, school, and study site. Study sites and schools nested within study sites were considered as having fixed effects. The denominator degrees of freedom for statistical tests pertaining to fixed effects were calculated using the Kenward and Roger approximation. These analyses were conducted using SAS version 9.3 (SAS Institute, Cary, NC, USA).

Curvilinear relationships of AST with distance to school and parent’s perception of crime were estimated using smooth terms in generalized additive models (GAMs) (in GAMs the linear predictor is specified in terms of a sum of smooth functions of covariates) [42]. We employed a GAM function of mgcv package in R with binomial variance with logit link function and used thin-plate regression splines to estimate the smooth function of the covariates distance to school and parent’s perception of crime. Separate GAMs were run to estimate the association of AST with distance to school and parent’s perception of crime by income level of the country that the study site belonged to. We used GAM to study the association of AST with distance to school and parent’s perception of crime because these models can estimate complex curvilinear relationships of unknown form among a dependent variable and smooth functions of a set of covariates and/or a set of covariates [42]. A detailed description of GAM is available elsewhere [42]. These analyses were conducted using R version 3.4.0 (The R Foundation for Statistical Computing, Vienna, Austria). 

The generalized linear mixed models and the GAMS were also conducted excluding the active commuters living at 5km or more from school as sensitivity analyses.

A distance decay parameter was estimated to compare the distribution of walking distances among study sites. A specific distance decay function fitted to a real data set presents a precise description of the distribution of walking trips over distances [43]. The exponential function is used because the distances involved are relatively short [44,45,46,47]. The function used is:(1)$$P\;\left(d\right)=e^{-\beta d}$$
where P(d) denotes the cumulative percentage of walking trips with distance equal or longer than d and β is the parameter estimated using empirical data. The parameter β was estimated by least-squares fit (FindFit in Mathematica 11.1). The resulting distance decay functions can be used to compare the distribution of walking distances among different groups [43]. 

### 2.5. AST Policies

To contextualize the policy environment of the study sites included in this analysis, we reviewed specific AST-policy documents at the city/state level. A policy search plan was developed to incorporate two different searching strategies: (1) academic databases and (2) customized Google search engines. The search strategy comprised four concepts: age group, active transport, interventions, and location of the interventions. These concepts were translated into keywords (adolescent, child, children, students, pupils, bicycling, transportation, walking, cyclists, cycling, bike, travel, intervention, implement, evaluate, change, pilot, project, environment, planning, impact, policy, project, politics, program, guidelines, methods, health impact assessment and planning techniques, Ottawa, Bogota, Helsinki, Cape Town, Baton Rouge, and Louisiana). For the customized Google search, we used the terms “Active school transportation policy + City/state” OR “City + school transport guide”, OR “City + school transport guidelines”; for Bogota, the search also included the Spanish terms: “*Bogotá guías de transporte escolar*”. The eligibility criteria included: impact evaluation of programs, case studies, policy documents, official guidelines, nongovernmental information, and news. The documents selected were screened for information on regulation of AST in each city (Appendix A). Finally, we extracted information from all of the documents regarding security, infrastructure, and the specific policy actions at each city. This information was complemented and validated by coauthors from each country site. 

## 3. Results

A total sample of 2845 children from Ottawa (n = 541), Bogota (n = 912), Helsinki (n = 487), Cape Town (n = 312), and Baton Rouge (n = 593) was included in the present analysis (Table 2). The average age of participants was 10.3 ± 0.6 years, and 54% were girls. Parental education level differed between sites, reflecting the variability in socioeconomic status. Cape Town had the highest percentage of parents with less than high school as their highest education level (37.7%), while Ottawa had the lowest percentage in this category (2.0%). Overall, 67.6% of the households had access to at least one vehicle. Ottawa had the lowest percentage of households with no access to motor vehicles (3.8%), while Bogotá had the highest percentage in this category (75.8%). Finally, the average score for crime perception was 2.6 ± 1.0, ranging from 1.6 in Helsinki to 3.4 in Bogota (Table 2).

### 3.1. School Transport

The overall prevalence of AST was 51.4%, ranging from 10.7% in Baton Rouge to 79.1% in Helsinki. Among all children, the average distance between home and school was 2.8 ± 4.0 km, ranging from 1.5 km in Helsinki to 4.6 km in Baton Rouge. Among children who engaged in AST, the average distance between home and school was 1.3 ± 2.4 km, ranging from 1.0 km in Helsinki and Baton Rouge, to 1.7 km in Cape Town. In the group of active travelers, 68.7% of the children lived within 1 km of the school, while 9.6% lived further than 2 km away (Table 2).

### 3.2. Factors Associated with AST by Income Level of the Country

Multivariable models stratified by income level of the country that the study sites belonged to showed common and differing factors associated to AST (Table 3). Number of vehicles and greater distance between home and school were negatively associated with AST in sites from both UMIC and HIC. In addition, children whose parents had a lower education level were more likely to engage in AST, only in sites from UMIC. Regarding crime perception, each unit increase in the crime perception scale was associated with 33% higher odds of AST among children from sites in UMIC (OR = 1.33 CI [1.06–1.66], *p* = 0.014), whereas an opposite association was observed among children from sites in HIC (OR= 0.37 CI [0.31–0.45], *p* < 0.001). Gender was not associated with AST and age was positively associated only among children from sites in HIC. The direction and significance of these associations remained in the sensitivity analysis excluding the children who lived at 5km or more and used AST (results not shown). 

Figure 1 shows the curvilinear relationship of AST with distance to school and parent’s perception of crime by groups according to the income level of the countries. The negative association between distance and the probability of engaging in AST was stronger in sites in HIC (Chi.sq (6.7, 8.4) = 423.5, *p* < 0.0001) compared to the sites in UMIC (Chi.sq (13.1, 16.2) = 332.6, *p* < 0.0001). In HIC-sites, the results of the GAM show that the probability of engaging in AST decreased with an increase of the distance to school from 0 to 5 km. However, for UMIC-sites, the probability of engaging in AST decreased but not uniformly when the distance increased. The probability for sites in UMIC increased again when the distance took the values of 5, 12, and 15 km approximately. It is important to note that the latter estimates had a high level of uncertainty (imprecise confidence intervals) due to the small number of participants living more than 15 km away from school. However, the analysis excluding the AST users living at 5km or more, showed similar patterns (results not shown).

Moreover, for sites in HIC, the results of the GAM show that the probability of engaging in AST decreased with an increase in the parent’s perception of crime from 1 to 3 (Chi.sq (3.9, 4.8) = 124.3, *p* < 0.0001). However, for sites in UMIC, the probability of AST increased with parent’s perception of crime (Chi.sq (3.3, 4.1) = 99.6, *p* < 0.0001).

Figure 2 shows the distance decay functions for each study site. This figure shows the lowest β-parameter for children from Cape Town (β = 0.87), followed by Ottawa (β = 1.05), Bogota (β = 1.15), Finland (β = 1.16), and the highest in Baton Rouge (β = 1.54). A higher β means a steeper decline in the probability of walking as distance increases.

### 3.3. AST Policies

Table 4 describes AST policies in the five cities included in this analysis. All the cities had AST-related policies, and all of them proposed actions aiming to change travel behavior and to create AST supporting environments. Transport and education sectors were the main leaders and implementers of these policies, but most of them also engaged other sectors like public health, urban planning, and security. A common characteristic of the policies was the inclusion of school travel initiatives such as Safe Routes to School (Ottawa and Baton Rouge), Al colegio en bici (Bogota), and walking school buses and bicycle trains initiatives (Helsinki and Cape Town). Reflecting the importance of distance as a determinant in the mode of transport selection, all the cities had programs or initiatives that enhanced public transportation or school buses for children living at a certain distance from the school. However, the eligibility criteria to access these programs differed by city. The minimum distance between home and school to be eligible for transport support varied from 0.8 km for kindergarten children in Ottawa to 5 km for children in Cape Town.

## 4. Discussion

The results of this study show a tale of two journeys, one in cities from UMIC and a different one in cities from HIC. In the study sites in UMIC, higher scores on crime perception were associated with higher odds of AST and there was a curvilinear relationship between distance and the likelihood of AST. Conversely, in the study sites in HIC, children were less likely to engage in AST as distance and crime perception increased. The fact that children from sites in UMIC are engaging in AST even with high perceptions of vulnerability to crime supports the previously proposed need-based framework for physical activity. This framework suggests that for some populations AST may be the result of a need in absence of other options for transportation [4,23]. By contrast, in high-income settings AST may be a socially desirable activity [60] in a supportive context [24], which supports a choice-based framework [4,23]. These differences in AST patterns should be considered for evaluating existing policy approaches and to support the development of new policy, regulation, design, and program interventions for children.

Safety concerns are one of the main barriers to AST reported by parents [22,61,62]. Our results for the sites in HIC are consistent with previous and recent evidence [63,64]. However, our findings indicate a counter-intuitive association among children from sites in UMIC. These results indicate that for many children, safety concerns are not a barrier for AST and suggest that these children could be engaging in AST due to a necessity. Our results contribute to fill the gap identified by recent studies in LMIC that make a call for the collection and report on cross-country differences in the drivers of AST that can be related to socioeconomic status, such as distance, car ownership, and safety [65]. Despite the fact that supporting evidence from other UMIC is scarce, similar associations have been reported in disadvantaged populations from HIC as observed in children from urban-dwellings in Baltimore [66]. Similarly, a longitudinal study on younger children in Quebec, Canada reported that those living in poverty were more likely to engage in AST during the first school years, despite being exposed to unsafe environments, which has been defined by the authors as environmental injustice [18]. These results add to the concept that in low income communities AST is a need instead of a choice. In addition, previous research in low income groups of adults [67] and children [68,69] has yielded conflicting findings regarding the relationship between crime and different measures of AST and PA. Our policy review evidenced that safety is mentioned as a priority in the agenda of AST promotion in our study sites. Interventions like Safe Routes to School, the Walking School Bus, or Al colegio en bici could serve as examples from cities in HIC and UMIC to improve safety and reinforce AST where it is already prevalent. These strategies highlight the importance of parents, school, and community involvement, as well as interaction between these groups. However, more evidence and impact evaluation of the different outcomes of these strategies are needed. Recent systematic reviews have reported the effectiveness of Safe Routes to School and Walking School Bus initiatives in other HIC contexts, but few interventions have been implemented in UMIC [50,70]. Future studies comparing multiple programs of UMIC and HIC settings should also be conducted.

The negative association with access to motorized vehicles observed in this study is consistent with previous literature showing that children from households with at least one motorized vehicle available for use are less likely to engage in AST [25,71,72]. However, when examining ISCOLE country sites individually, this relationship was not significant in many country sites [25]. Presumably, vehicle ownership may favor motorized travel to a greater extent when parents perceive that it is more convenient to drive their children to school [73,74]. In North America, AST has declined considerably over the last 50 years [17,21,22]. For example, in the United States, 49% of children 5-14 years engaged in AST in 1969, but only 13% engaged in AST in 2009 [22]. These trends and our findings suggest that in HIC settings, interventions should focus on children from households that own a car and live at a walkable distance from school, which could potentially shift from motorized to active modes of transport. Our study shows that 73.3% of children in Bogota, 79.1% in Helsinki, and 50.3% in Cape Town engage in AST, which reflect even higher AST patterns than those observed in the 1960s in the United States. Future studies should assess the factors associated with AST in a HIC setting like Finland and in UMIC like Colombia and South Africa. To our knowledge, no previous study in LMICs examined perceived convenience of driving among parents or to what extent the (re)development of built environments designed to prioritize cars could or has affected AST, as it has been observed in HIC [61]. Such research may be particularly important in the context of the PA transition [75], which is characterized among other things by a shift from active to motorized transportation. 

The negative association observed between distance to school and AST in HIC and in UMIC is consistent with previous studies that have reported that distance is one of the main correlates of AST [26,27,30,64,71,76,77]. However, our results show different parameters and inflection points by study site and by income level of the country, respectively. Specifically, the distance decay parameters indicate that children from Cape Town and Ottawa are willing to actively travel for longer distances. Nonetheless it is in Cape Town and Bogota where the percentage of children have the highest likelihood of walking more than 5 km. These results align with the proposed need vs. choice framework [4,23]. Similarly, the curvilinear relationship between AST and distance observed for sites in UMIC suggests an increased probability of engaging in AST among children living 5 km or further from the school. Considering that previous studies in HIC have proposed distances between 1.4 and 1.6 km as thresholds for socially normative walking distances for children [78], we hypothesize that our results for sites in UMIC could be related to poverty conditions. Children who are walking those extreme distances because they have no other choice, usually face risks and challenges that eventually would lead them to give up active commuting [60]. For this reason, these groups should be the target for cycling or multimodal transportation initiatives that improve their quality of life while at the same time allowing them to maintain active behaviors for an acceptable part of their journey. Initiatives that provide the bicycles to children through a loan system, like Al colegio en bici in Bogota and Qubeka in Cape Town could be scalable to cities in LMIC to provide access to the required equipment for cycling to school. The policy review indicated that all the study sites are committed to provide school bus services or subsidies for children living at certain distances from school; however, our results could suggest a limited implementation of these initiatives in the sites in UMIC. The findings from this study could contribute to understand the role of urban planning and safety promotion at the local level in the engagement on AST. These associations are relevant for the promotion of health in urban contexts, and increasingly car-dependent societies, taking into account the multiple health and environmental benefits of active transportation [79].

Strengths of the present study include the implementation of a standardized protocol across countries which facilitated comparisons, the objective measurement of distance between households and schools and the use of a crime subscale with a satisfactory internal consistency, and the systematic search of local policies. However, our findings should be interpreted cautiously considering the following limitations. First, the cross-sectional design of the study does not allow the inference of causality. For instance, we cannot rule out the possibility that the observed relationship between perceived crime and AST might be attributable to reverse causality (i.e., parents may be more concerned about crime because their child engages in AST). Second, AST was assessed only for the journey “to school”, which may be a potential source of bias, considering that some children may engage in AST only on the journey “to home”. Third, the distance estimations assumed that children took the shortest route via the street network to go to school, which may not represent the actual route travelled [26,80]. However, a previous study that compared routes measured by global positioning systems and estimated by GIS found similar travel distances, despite the route discordance [81]. Fourth, correlates of walking and cycling may differ but the low prevalence of cycling in all the countries, except for Finland, did not allow for the assessment of these relationships separately. Fifth, there could be a risk of recall bias and social desirability bias in the AST and parental perception of crime variables, due to the self-reported nature of these variables. Finally, sixth, regarding the results for UMIC, it is important to note the lower inclusion rate of participants from Cape Town, which led to having an overrepresentation of children from Bogotá in the UMIC subsample.

### Policy Implications

Our results can be of interest for policy-makers from multiple sectors in high-income countries and UMIC. Ideally, these findings should be taken into account in conjunction, as evidence suggests that the most promising policies are those that address distance concerns and safety perceptions improvements [82]. The following points summarize the relevance of our findings for policy-design in different contexts:Distance: Distance is a key determinant for school-siting policies that aim to create dense school networks and encourage active transportation by the location of schools at reasonable distances from residential neighborhoods. Through the design of school-siting and land use policies, policy-makers can manage the strong influence that distance can have on the engagement in AST. This could be a relevant strategy for settings that are experiencing urbanization and growing processes. Extreme distances can be addressed by multimodal transportation strategies that combine motorized travel and AST, such as bike-friendly features in public transportation infrastructure or safe routes for walking from bus stops to schools.Safety: policy-makers can contribute to address the parents’ concerns about crime perceptions through the identification of potential risks and the design of safe routes for children living at walkable distances from the schools. Strategies that involve adult accompaniment along the trip to school can encourage the involvement in AST. These strategies are pertinent for HIC settings where safety is inversely related to AST and for UMIC settings where an inverse association was not observed, but safety improvement can contribute to making AST a sustainable behavior. Car-ownership: policy-makers cannot restrict car-ownership, however, motorized transportation can be made less convenient for short trips through policy. Initiatives such as parking restrictions around schools and traffic management strategies can discourage the use of motorized modes and replace those trips with active options. 

## 5. Conclusions

Distance to school is a consistent correlate of AST in differing contexts. Our findings regarding crime perception support the need vs. choice framework, indicating that AST may be the only travel mode available for many children from UMIC settings, despite the high perception of crime. These findings could contribute to the design of policies and programs intended to promote active commuting among children. The observed differences in the correlates of AST by country-income level further substantiate our previous recommendation for context-specific evidence to guide local interventions [25]. Future studies to address models of transportation to school behaviors should be conducted taking into account the local context of the studied area and the potential differences within and between regions. Policies and programs should be implemented to promote AST and ensure that safe routes are available with the goal of reversing the declining trend of AST in HIC and to maintain and increase prevalence of AST in UMIC settings, before unintended consequences of development change these patterns.

## Figures and Tables

**Figure 1 ijerph-17-03847-f001:**
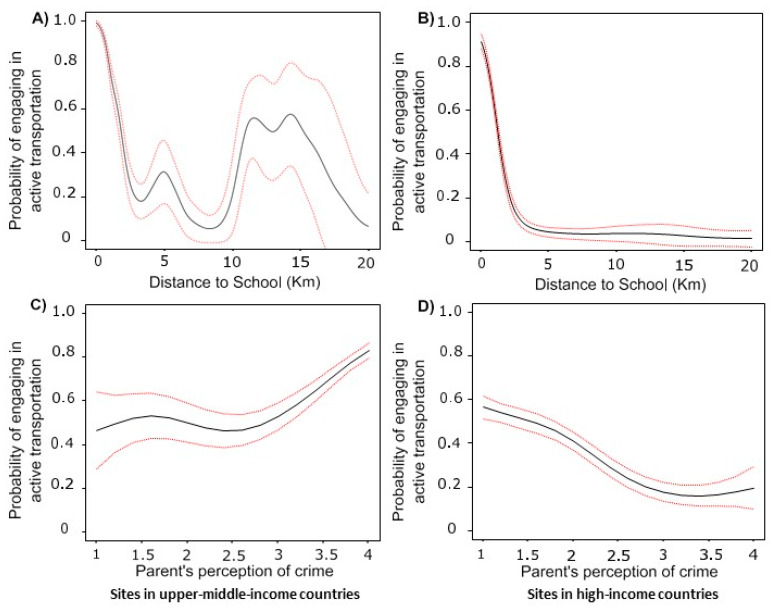
Associations of active transport to school with distance and crime perception by income level of the country. (**A**) Association of active transportation to school with distance between home and school among children from sites in upper middle-income countries. (**B**) Association of active transportation to school with distance between home and school among children from sites in high-income countries. (**C**) Association of active transportation to school with parental perception of crime in sites in upper middle-income countries. (**D**) Association of active transportation to school with parental perception of crime in sites in high-income countries.

**Figure 2 ijerph-17-03847-f002:**
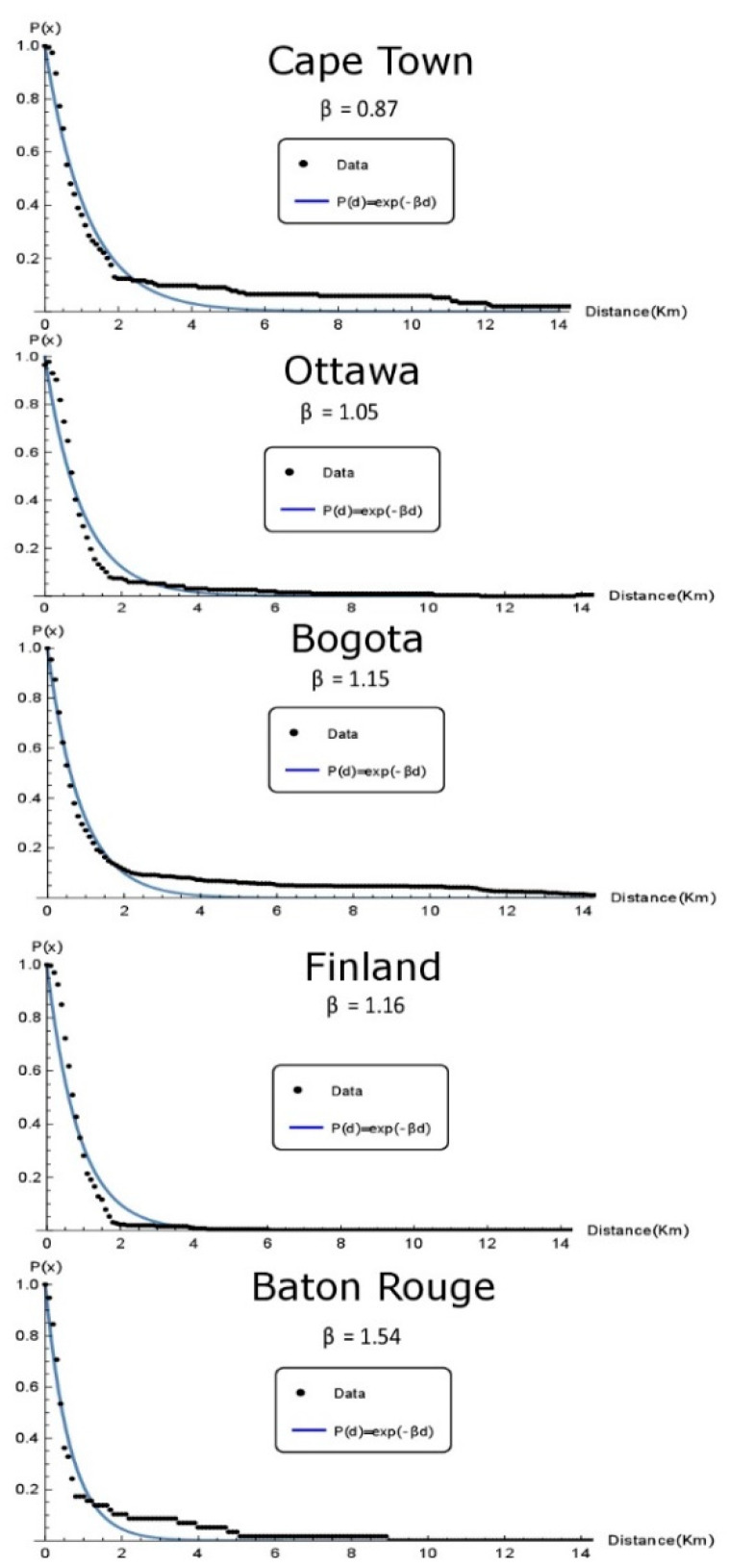
Distance decay curves by study site.

**Table 1 ijerph-17-03847-t001:** Sociodemographic characteristics of five country sites in the International Study of Childhood Obesity, Lifestyle and the Environment (ISCOLE).

Socio-Demographic Characteristics	Ottawa (Canada)	Bogota (Colombia)	Helsinki, Espoo & Vantaa (Finland)	Cape Town (South Africa)	Baton Rouge (US)
World bank classification ^a^	High income	Upper-middle income	High income	Upper-middle income	High income
Gini index (year) ^a^	34.0 (2013)	50.8 (2016)	27.1 (2017)	63.0 (2014)	41.5 (2016)
Total population at the city level	812,129	7,674,366	1,005,275	3,433,441	228,590
Population density (inhabitants per km^2^)	317	4310	2739	1530	2960
Motor vehicles per 1000 inhabitants ^b^	605	58	534	159	809
Estimated road traffic death rate per 100,000 population ^c^	6.8	15.6	5.1	31.9	11.4
Crime rate					
Robbery rate per 100,000 population ^d^	58.8	197.5	30.8	101.4	102

^a^ World Bank Data at country level [33]; ^b^ World Bank Data at country level: Motor vehicles (per 1000 people) include cars, buses, and freight vehicles but not two-wheelers [35]; ^c^ World Health Organization data at country level: Global status report on road safety 2013 [36]; ^d^ Robbery at the national level, number of police-recorded offences. Definitions: “Robbery” means the theft of property from a person, overcoming resistance by force or threat of force. Where possible, the category “Robbery” should include muggings (bag-snatching) and theft with violence but should exclude pick pocketing and extortion [37].

**Table 2 ijerph-17-03847-t002:** Descriptive Characteristics of Participants Stratified by Study Site (n = 2845) in the International Study of Childhood Obesity, Lifestyle and the Environment (ISCOLE).

Socio-Demographic Variables of the Sample	Ottawa (Canada)	Bogota (Colombia)	Helsinki, Espoo & Vantaa (Finland)	Cape Town (South Africa)	Baton Rouge (US)	Total
	*n* = 541	*n* = 912	*n* = 487	*n* = 312	*n* = 593	*n* = 2845
Age ^a^	10.5 (0.4)	10.5 (0.6)	10.5 (0.4)	10.2 (0.7)	10.0 (0.6)	10.3 (0.6)
Sex						
Male (%)	42.7	49.6	47.5	44.3	43.2	46.0
Female (%)	57.4	50.4	52.5	55.7	56.8	54.0
Highest parent education						
<High School (%)	2.0	31.8	2.9	37.7	8.6	17.0
Complete high-school or some college (%)	27.8	50.8	54.9	45.9	43.2	45.0
≥Bachelor degree (%)	70.2	17.4	42.2	16.4	48.2	38.0
Number of motorized vehicles in the household						
None (%)	3.8	75.8	9.4	37.5	8.3	32.5
One (%)	38.3	21.5	45.2	32.4	30.5	31.8
Two or more (%)	57.9	2.7	45.4	30.1	61.2	35.7
Crime perception score ^a^	2.0 (0.7)	3.4 (0.7)	1.6 (0.6)	3.1 (0.8)	2.4 (0.8)	2.6 (1.0)
**School transport characteristics**						
Mode of transport to school						
Walking (%)	34.9	71.6	54.7	49.4	10.1	46.3
Bicycle, roller-blade, skateboard, scooter (%)	0.6	1.8	24.4	0.9	0.7	5.1
Bus, train, tram, underground, or boat (%)	38.1	18.7	13.3	5.4	34.5	23.3
Car, motorcycle, or moped (%)	26.5	7.3	7.6	44.3	54.3	25.0
Other ^b^ (%)	0.0	0.7	0.0	0.0	0.5	0.3
**Distance-related variables**						
Average distance to school (km) ^a^	2.8 (4.2)	2.4 (3.7)	1.5 (1.7)	2.9 (3.9)	4.6 (5.1)	2.8 (4.0)
Median of the distance to school (km)	1.5	0.8	1.0	1.5	3.3	1.3
Distance distribution among active and nonactive travelers						
< 1 km (%)	36.8	56.6	50.0	38.1	19.1	41.7
1 km ≤ Distance < 1.5 Km (%)	13.2	10.6	20.9	12.0	11.6	13.2
1.5 Km ≤ Distance < 2 Km (%)	10.9	5.5	12.3	11.3	7.9	8.8
≥2 km (%)	39.2	27.4	16.8	38.7	61.4	36.2
Distance distribution among active travelers						
< 1 km (%)	70.0	73.0	60.6	64.4	80.0	68.7
1 km ≤ Distance < 1.5 Km (%)	17.1	10.6	21.8	12.5	4.6	14.3
1.5 Km ≤ Distance < 2 Km (%)	4.2	4.9	12.2	11.3	3.1	7.3
≥2 km (%)	8.8	11.5	5.4	11.9	12.3	9.6
Average distance to school among active travelers (km) ^a^	1.3 (2.9)	1.4 (2.7)	1.0 (0.8)	1.7 (3.2)	1.0 (1.6)	1.3 (2.4)
Active travel among children living at <1 km (%)	67.2	94.6	95.9	84.9	44.8	84.5

^a^ Mean and Standard Deviation; ^b^ Other includes school van, bus feeder, riding on the top tube of the bike’s frame, pedicab, and wheelchair.

**Table 3 ijerph-17-03847-t003:** Factors associated to active school transport in 2845 9–11-year-old children, by income level of the country.

Covariates	Sites in Upper-Middle-Income Countries ^a^			Sites in High-Income Countries ^b^		
	OR	95% CI	*p*-Value	OR	95% CI	*p*-Value
Highest parent education						
<High School	4.83	(2.84–8.21)	< 0.001	0.89	(0.45–1.78)	0.741
Complete high-school or some college	4.21	(2.58–6.85)	< 0.001	1.35	(1.01–1.81)	0.040
≥Bachelor degree	Ref.			Ref.		
Age	0.80	(0.61–1.06)	0.126	1.96	(1.49–2.58)	< 0.001
Gender (ref. male)	1.27	(0.89–1.79)	0.176	0.97	(0.74–1.28)	0.838
Crime perception	1.33	(1.06–1.66)	0.014	0.37	(0.31–0.45)	< 0.001
Number of motorized vehicles (ref. none)						
None	Ref.			Ref.		
One	0.24	(0.16–0.35)	< 0.001	0.42	(0.24–0.72)	0.002
Two or more	0.14	(0.08–0.26)	< 0.001	0.38	(0.22-0.65)	0.001
Distance to school						
<1 Km	Ref.			Ref.		
1 km ≤ Distance < 1.5 Km (%)	0.12	(0.07–0.20)	< 0.001	0.29	(0.21–0.42)	< 0.001
1.5 Km ≤ Distance < 2 Km (%)	0.13	(0.07–0.23)	< 0.001	0.15	(0.10–0.22)	< 0.001
≥2 Km	0.03	(0.02–0.05)	< 0.001	0.02	(0.02–0.03)	< 0.001

^a^ Sites in upper-middle-income countries comprised Bogota and Cape Town according to the World Bank classification [35]; ^b^ Sites in high-income countries comprised Ottawa, Helsinki and Baton Rouge according to the World Bank classification [35].

**Table 4 ijerph-17-03847-t004:** Description of policies that support active school transport in Ottawa, Bogota, Helsinki, Cape Town, and Baton Rouge.

Location	Description	Target	Sectors Involved	Impact Evaluation
Ottawa, Canada	The Ottawa Student Transportation Authority (OSTA) is responsible for all school transport initiatives and policies at the city. Regarding active school transportation (AST), OSTA provides services that support and promote the core principles of the School Active Transportation Charter. Specific actions include: (1) Assisting schools in providing safety conditions for students through management of vehicle, pedestrian, and bike traffic around schools. (2) Assessing potential hazards in all walk zones and assigning transportation services to those children who walk and face a very high risk to their safety. (3) Recommending the best routes for AST, through maps that identify unsafe intersections to avoid. (4) Submitting infrastructure improvement needs or service requirements to the appropriate departments at the city. (5) Coordinating School Travel Planning initiatives, like Active and Safe Routes to School program, that involve school communities engaged in the development of action plans for removing barriers to AST. (6) Coordinating Walking School Bus initiatives, in which children are encouraged to walk to school accompanied by a paid leader of the program. AST programs and policies are also supported by the Ottawa School Active Transportation Network, which involves OSTA, School Boards planning, Ottawa Police Services, City By-Law, Ottawa Public Health, Ottawa Public Works, Green Communities Canada, and Ottawa Safety Council [48]. School board policies determine the eligibility for bus services based on the distance between home and school as follows: kindergarten students located at ≥ 0,8 km, grades 1 to 8 located at ≥ 1.6 km, and grades 9 to 12 located at ≥ 3.2 km or more from their home school [48].	Parents or guardians, students, school communities	OSTA, School Boards planning, Ottawa Police Services, City By-Law, Ottawa Public Health, Ottawa Public Works, Green Communities Canada, and Ottawa Safety Council	School Travel Planning initiatives have been evaluated in Canada. Mammen et al. reported pooled data from several cities across Canada, but no specific data was provided for Ottawa. This evaluation found that after 1 year of implementation, there was no increase in AST. However, given the school-specific nature of the program, this approach may not be appropriate to evaluate its impact [49,50].
Bogotá, Colombia	The main AST policy in Bogotá is the School Mobility Plan, which was designed and enforced by the School Board and the District Department of Transport. This plan comprises guidelines for motorized and nonmotorized school transport. Each school must design its own Mobility Plan and propose strategies to promote active and sustainable mobility. The specific actions of this policy regarding AST include: (1) Assigning children to the closest schools to their homes, in order to promote active commuting. (2) Improving infrastructure prioritizing safety conditions for pedestrians and cyclists. (3) Implementing programs to promote safe walks to school among students living at 2 km or closer. (4) Implementing the program "*Al Colegio en Bici*", a comprehensive program to promote biking in public schools that includes a bicycle loan system, supervision, and education strategies. The education strategies comprise training in road safety, cycling skills, traffic rules, and a participatory design of safe routes [51]. Based on the distance between home and school, and considering vulnerability of children, bus services, or transport subsidies (money transfers or bus card) can be provided. Eligibility for motorized transportation benefits is defined as follows: ≥ 1 km for kindergarten children and ≥ 2 km for children from 1st to 11th grade.	Parents, community, students	Education, planning, mobility, sport and recreation, urban development, security road and maintenance and security department [51].	No impact evaluation
Helsinki, Finland	Helsinki Region Transportation (HRT) is the main authority in charge of the transport policy and mobility plans. The main policy document to guide specific actions to promote AST in Helsinki is the School Mobility Plan [52]. Each school is independent in the design and development of their mobility plan. However, the common purpose is to increase the use of walking, cycling, or public transport and make commitments in sustainable practices. Mobility plans also aim to increase the independent mobility among schoolchildren, as well as the safety in walking and cycling trails. Overall, the plan should include (1) identifying mobility problems or characteristics of the environment that make active commuting to the school difficult. (2) Assigning responsible persons for the implementation, including the school principal. (3) Formulating a mobility study at the school level. (4) Establishing objectives and achievable goals for the plan. And (5) Proposing an action plan, with specific initiatives like walking or cycling school buses, mobility lessons, or bicycle service days at school. Public transportation is encouraged through the entitlement of a Helsinki Region Transport travel card for children living at further distances based on the following criteria: children from 1st to 6th grades with journeys ≥ 2 km or adolescents from 7th to 9th grade with journeys ≥ 3km. For shorter distances, the use of the journey planner for cycling and walking is encouraged [52,53].	Parents or guardians, students, school communities	Transport and education	No impact evaluation
Cape Town, South Africa	Cape Town’s Transport and Urban Development Authority (TDA) is responsible for the local transport policies. The main policy to promote AST in the city is the Non-Motorized Transport (NMT) Policy and Strategy. This document aims to create safe environments for pedestrian and cyclists in order to increase AST as a desirable and acceptable mode of transport. Specific actions related to schools and learners in this policy include: (1) promoting scholar patrols, (2) implementing bicycle/pedestrian paths and other NMT infrastructure in school priority zones, (3) introducing walking and cycling bus programs and (4) including learner safety programs as part of the school curriculum [54]. However, is important to highlight that the school transport policies are focused on the provision of motorized school transportation based on the extreme distances that most of the children walk to school and challenges that children face on their way to and from school [55,56]. Specific criteria for Western Cape province establishes that children who live at 5km or further from their school are eligible for school transport provision [55].	Community, Students	Transport, education, and urban development	No impact evaluation
Baton Rouge, U.S.A	The Louisiana Department of Education regulates the School Transportation for Louisiana. The main guidelines for school transportation are provided on the Louisiana School Transportation Specifications and Procedures Bulletin. However, this document is focused in motorized transportation to school and specifies that children whose home is located further than 1 mile from the school should be provided with free transportation, and children living within 1 mile can also be eligible for bus transportation in case of hazardous walking situations [57]. A more supportive policy for AST is the Complete Streets policy at the state level in Louisiana, led by the Louisiana Department of Transportation and Development (DOTD). This policy aims to create an integrated transportation network that provides access, mobility, and safety to the users of different transport modes, including active modes, in Louisiana [58]. One of the expected benefits of the implementation of this policy is increased road safety and active transportation among children. In partnership with the DOTD, school travel initiatives like Safe Routes to School (SRTS) are funded with the objective to improve the health of kids and the community by making walking and bicycling to school safer, easier, and more enjoyable. SRTS programs comprise five components: education, encouragement, enforcement, engineering and evaluation, and include specific actions, such as: (1)Teaching safety skills, (2) Creating awareness among students, pedestrians, and bicyclists, (3) Helping children to follow transit rules and (4) Improving driver behaviors [59].	Parents, Students, Schools	Transport, education, and planning	Safe Routes to School impact has been evaluated but not in Louisiana

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
