# Peer review of "Active School Transport among Children from Canada, Colombia, Finland, South Africa, and the United States: A Tale of Two Journeys"

_ijerph, 2020, doi:10.3390/ijerph17113847_

Round 1

Reviewer 1 Report

  • Have the authors thought of multiple imputations to prevent loss due to missing data?
  • The inclusion rate in Bogota is a low lower than the rest of the sites. This should be addressed in the discussion.
  • How did the authors take into account the clustering within schools? Multilevel models were conducted but could the authors be more clear if these models were 2 level (students nested within schools) or 3 level (students nested within schools nested within sites)
  • Could the authors include more objective measures of safety, such as crime rate, pedestrian-vehicle collisions, etc.
  • The authors observed crime was associated with AST only in sites within high-income countries, not middle-income countries. So, in other words, those living in high-income countries have a choice not to use AST in perceived high crime sites. However, according to this paper in Montreal Canada,

  • Pabayo R., Gauvin L., Barnett TA., Morency P., Nikiéma B., Seguin L.  Understanding determinants of active transportation to school among children living in poverty: Evidence of environmental injustice from the Quebec Longitudinal Study of Child Development. Health Place 2012 Mar;18(2):163-71. Epub 2011 Sep 10.

  • These authors found a similar association with those in Middle income countries. Those from lower SES backgrounds did not have a choice and were more likely to use AST within areas with high pedestrian-vehicle collision rates.

Author Response

We appreciate the reviews received. Please see the attachment to find our point-by-point responses. 

Reviewer 2 Report

This is a well written manuscript that has used appropriate methods. My only concern with the study is generalising the results to MIC given that the World Bank has classified Bogota and Captown as Upper-Middle income. I would like the author to not this in the limitations section of the manucript

I would also like the author to remove the reference to LMIC in line 66 of the manuscript as this is not the focus of this research.  

Author Response

(The authors gave the same response as above.)

Reviewer 3 Report

The paper deals with an important problem concerning children's physical activity and how they go to school. As part of the study, five countries were divided into two groups: middle- and high-income countries. The research concerned the impact of selected factors on the participation of active school transport (AST) in selected countries. Action strategies to increase the share of AST in the countries analyzed were also compared.

The aim of the research is clearly stated.

But why are the results of the ISOLE project from 2013 only now being published? Wouldn't it be worth making a comparison of what has changed in recent years, especially in middle-income countries?

Table 1 indicates the road safety indicator, in the further part of the study there is no reference to this factor. Hasn't the impact of road safety been tested on the children’s way to school? It seems that this can be a very important factor - assessment of this safety based on accident data as well as a sense of safety for children and parents. The safety of children in road traffic can be more important than crime perception and decide how children go to school.

Selected factors are given for the entire country - Table 1, while there potentially could be differences when taking into account, for example, a specific city.

I see no point in having table 4.

The authors rightly admit that the obtained results cannot be treated as explaining the whole of this complex phenomenon. However, such research is needed, it should be developed, building models of behaviour, while differentiating the studied areas. In high-income countries, there can be very large differences, larger than the authors pointed out, for instance in the EU itself there will be very large differences.

I assess the whole as an interesting contribution to the recognition of the problem of physical activity of children, it is a pity that the data is not up to date.

Author Response

(The authors gave the same response as above.)
